# SqueezeAttention: 2D Management of KV-Cache in LLM Inference via Layer-wise Optimal Budget

Zihao Wang,[*] Bin Cui[†], Shaoduo Gan[†]

School of CS & Key Lab of High Confidence Software Technologies (MOE), Peking University
Geoming AI
hetailang0.o@gmail.com, {bin.cui, shaoduo.gan}@pku.edu.cn

## Abstract

Optimizing the Key-Value (KV) cache of the Large Language Model (LLM) has been considered critical to saving the cost of inference. Most of the existing KV-cache compression algorithms attempted to sparsify the sequence of tokens by taking advantage of the different importance of tokens. However, most of these methods treat all layers equally, allocating the same KV budget to each layer. This approach is suboptimal, as some layers may be less sensitive to input tokens yet still receive the same budget as others. In this work, we found that by identifying the importance of attention layers, we could optimize the KV-cache jointly from two dimensions, i.e., sequence-wise and layer-wise. Based on our observations regarding layer-wise importance in inference, we propose SqueezeAttention to precisely optimize the allocation of KV-cache budget among layers on-the-fly and then incorporate three representative sequence-wise algorithms to compress the KV-cache for each layer with its very own budget. Specifically, we first measure each layer's importance by calculating the cosine similarity of the input prompt differences before and after the self-attention layers. Based on this similarity, we then categorize the layers into two groups and adjust their KV budgets accordingly. By optimizing the KV-cache from both sequence's and layer's dimensions, SqueezeAttention achieves around 30% to 70% of the memory reductions and up to 2.2 × of throughput improvements in a wide range of LLMs and benchmarks. The code is available at https://github.com/hetailang/SqueezeAttention.

## 1 Introduction

The remarkable performance achieved by generative large language models (LLM) across a wide range of natural language processing (NLP) tasks is making people in the computing industry believe that it has a great potential to reshape the way they design their products. The past year has witnessed an unprecedented surge in applications driven by LLMs, such as intelligent chatbots, LLM-powered search engines, digital personal assistants, automatic programming tools, and so on. Along with the ever-growing LLM applications, their massive inference cost starts becoming a severe challenge that hinders the deployment of LLMs and raises concerns regarding their carbon footprint Faiz et al. (2023).

For a decoder-only autoregressive model, which is the most widely adopted LLM architecture, the inefficiencies in inference mainly come from the fact that the model can only generate tokens one by one, and sampling each token requires attending to all previous tokens. In practice, the intermediate key-value embeddings of each layer have been cached incrementally in each iteration to avoid frequent recomputations. Since the KV-cache increases linearly with the **number of attention layers**, **context length** and **batch size**, it often ends up being multiple times larger than the model itself

---

[*]This work is done during the internship at Peking University and Geoming AI
[†]Corresponding author

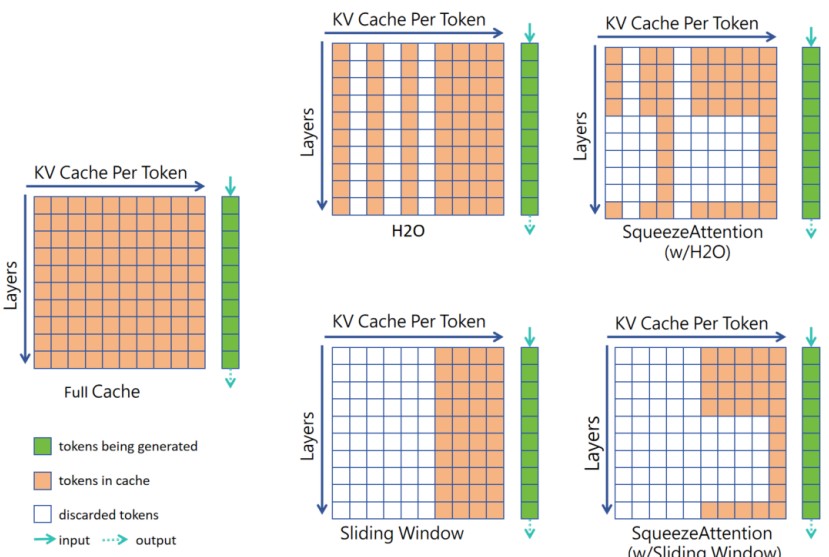

Figure 1: Demonstrations of KV-cache policies in inference from the view of the sequence and attention layer. Full Cache (leftmost column) simply stores the KV embeddings for all the tokens in all the layers. Sequence-wise compression algorithms (middle column) drop tokens in the sequence's dimension, where each layer has the same cache budget. SQUEEZEATTENTION (rightmost column) further compresses the KV-cache by adaptively re-allocating the cache budgets in the layer's dimension.

Sheng et al. (2023), and therefore, dominating the I/O cost of inference. Recently, optimizing the KV-cache has been broadly considered a critical approach to boost the efficiency of inference. From the perspective of context length, many well-studied algorithms are trying to identify the most "valuable" tokens in the sequence and evict the unimportant ones to reduce the KV-cache and attention complexity, such as Sliding Window Attention Beltagy et al. (2020), Heavy-Hitter (H2O) Zhang et al. (2024), StreamingLLM Xiao et al. (2023), Scissorhands Liu et al. (2024), FastGen Ge et al. (2023) and so on. From the perspective of batching, many studies aim to explore how to efficiently manage the memory of KV-cache on a batch basis with different sequence lengths Zheng et al. (2023); Kwon et al. (2023). However, the opportunities in the dimension of attention layers have barely been touched by most, if not all, of the existing methods. In other words, all the attention layers have always been treated equally by those KV caching strategies. Therefore, in this paper we ask:

*Do all the attention layers that share the same KV caching strategy have to cache the same amount of tokens? If not, how can we precisely allocate the cache budget for each layer such that we can further reduce the KV-cache on top of sequence-wise compressions?*

To answer these questions, we need to take a closer look at the behaviors of different attention layers during inference. Some inspiring clues could be found in a few existing studies. Early-exiting LLM Del Corro et al. (2023), as a widely-adopted inference method, shows that after going through a certain number of attention layers, the hidden representations are likely to reach saturation, and therefore, the forward computing can exit early without finishing the entire network and still get a reasonable prediction. Besides, a very recent work called FastGen Ge et al. (2023) found that attention layers in different positions have different optimal KV caching strategies. For example, attention layers in the very early part of the model should simply cache all the tokens in the sequence, whereas, some middle layers should apply cache eviction strategies based on the token locality or frequency. Although FastGen could select the optimal eviction strategy for each layer, it is still unclear how to find the optimal cache budgets, instead of a pre-defined, unified hyperparameter, for layers that share the same strategy.

Given the fact that attention layers do have different degrees of importance regarding inference, we can make a reasonable hypothesis that by taking advantage of the layer importance, we could

further "squeeze" the amount of KV-cache that has already been compressed by those sequence-wise eviction algorithms, and eventually, achieve even better efficiencies.

To describe the importance of the attention layers quantitatively, we track the cosine similarity, which has been considered a robust metric to reflect the similarity of embeddings in NLP Sidorov et al. (2014), between the hidden representations before and after the self-attention computing in each layer, and then put all layers' data together to demonstrate how an input embedding evolves through the entire model in inference. The intuition is that the more similar the embeddings are after the attention computing (indicated by higher cosine similarity), the less information this attention layer could insert into the embedding. After broad investigations into multiple popular LLM models, e.g., Mistral-7B, Falcon-7B, Llama2-7B, Llama2-70B, and so on, as shown in Figure 2, we found some common characteristics. Firstly, the first half of attention layers, in general, contributes more to the output embedding than what second half does. Secondly, some specific layers, typically the first and last few layers, might be more important than other layers, depending on the specific model and dataset.

Based on this simple yet effective indicator, we propose SQUEEZEATTENTION, a 2D KV-cache compression algorithm that prunes KV-cache from not only the sequence's dimension but also the layer's dimension. Since the layer importance is highly dependent on the model and task, SQUEEZEATTENTION categorizes all the layers into groups on the fly by clustering their cosine similarities measured during the prompt prefilling phase. Given a sequence-wise KV-cache eviction policy (like Sliding Window Beltagy et al. (2020) or H2O Zhang et al. (2024)), and a unified cache budget (like 4096 tokens or 20% of prompt length), SQUEEZEATTENTION automatically reallocates the cache budgets among groups of layers such that the important layers could cache more tokens to stabilize the model accuracy and the unimportant layers could drop more tokens to save the I/O cost. What's even better is that SQUEEZEATTENTION is orthogonal to all those sequence-wise KV-cache compression algorithms, so it can be smoothly combined with any of them. Figure 1 demonstrates how the SQUEEZEATTENTION works jointly with two representative sequence-wise KV-cache eviction algorithms, i.e., H2O Zhang et al. (2024) and Sliding Window Attention Beltagy et al. (2020). More details about the SQUEEZEATTENTION algorithm can be found in Section 4.

To the best of our knowledge, SQUEEZEATTENTION is the first algorithm considering the KV-cache budget in a layer-wise way, making it a valuable addition to all those sequence-wise compression algorithms for inference. In our experiments, we integrate SQUEEZEATTENTION into 7 popular LLM models ranging from 7B to 70B, i.e., Llama2-7B, Mistral-7B, Falcon-7B, OPT-6.7B, GPT-Neox-20B, Mixtral-8×7B, and Llama2-70B, combining with 3 representative sequence-wise KV-cache compression algorithms, i.e., Heavy-Hitter Oracle (H2O), Sliding Window Attention and StreamingLLM. The results show that SQUEEZEATTENTION can achieve better model performance with even lower cache budgets than all three algorithms under a wide range of models and tasks, which lead to approximately 30% to 70% of the memory savings and up to 2.2 × of throughput improvements for inference.

## 2 PRELIMINARIES AND RELATED WORK

### 2.1 ANATOMY OF KV-CACHE IN LLM INFERENCE

For a decoder-only transformer-based model, the inference process typically involves two phases: **prefilling** and **decoding**. In prefilling, LLM takes the entire prompt as input to calculate and cache the key-value embeddings of each token in each attention layer. Then the decoding phase takes one embedding at a time to generate tokens by iterations, and meanwhile, concatenates the newly calculated KV embedding to the KV-cache.

Let $p$ be the length of the prompt, $o$ be the length of the output, $b$ be the batch size, $n_{layer}$ be the total number of attention layers, and $d_{model}$ be the hidden dimension. In the $i$-th layer, denote the model weights regarding attention *Key* and *Value* by $\mathbf{w}_K^i$ and $\mathbf{w}_V^i$, where $\mathbf{w}_K^i \in \mathbb{R}^{d_{model} \times d_{model}}$, and $\mathbf{w}_V^i \in \mathbb{R}^{d_{model} \times d_{model}}$.

In the prefilling phase, denote the hidden states of the $i$-th layer by $\mathbf{h}_{prompt}^i$, where $\mathbf{h}_{prompt}^i \in \mathbb{R}^{b \times p \times d_{model}}$. Then the KV-cache of the $i$-th layer after prefilling can be formulated as:

$$\mathbf{C}_K^i = \mathbf{h}_{prompt}^i \cdot \mathbf{w}_K^i ; \mathbf{C}_V^i = \mathbf{h}_{prompt}^i \cdot \mathbf{w}_V^i \tag{1}$$

In the decoding phase, denote the hidden states of the $j$-th output token in $i$-th layer by $\mathbf{h}_{output_j}^i$ ($1 \leqslant j \leqslant o$), where $\mathbf{h}_{output_j}^i \in \mathbb{R}^{b \times 1 \times d_{model}}$. Then the KV-cache of the $i$-th layer after generating $j$-th tokens can be formulated as:

$$\mathbf{C}_K^i = \text{CONCAT}(\mathbf{C}_K^i, \mathbf{h}_{output_j}^i \cdot \mathbf{w}_K^i) \tag{2}$$

$$\mathbf{C}_V^i = \text{CONCAT}(\mathbf{C}_V^i, \mathbf{h}_{output_j}^i \cdot \mathbf{w}_V^i) \tag{3}$$

As the decoding process goes along, KV-cache is growing incrementally until the output sequence is fully finished. Therefore, the maximum number of floats in total of the KV-cache is:

$$2 \cdot \sum_{i=1}^{n_{layer}} [b \cdot (p + o) \cdot d_{model}] \tag{4}$$

or simply $2 \cdot d_{model} \cdot n_{layer} \cdot b \cdot (p + o)$.

Taking Llama-2-7B in FP16 as an example, where $n_{layer} = 32$, $d_{model} = 4096$. The entire model weights consumes around 14GB of memory, whereas, the KV-cache takes around 0.5MB per token. In other words, the KV-cache starts to exceed model weights when processing more than 28K tokens, which could be easily made up of a batch of 28 requests with 1K content length.

## 2.2 EXISTING KV-CACHE OPTIMIZATIONS

As analyzed above, the number of layers, batch size, and context length are three critical factors that decide the size of the KV-cache. Therefore, existing optimization studies are likely to seek opportunities from these perspectives.

Sparsifying the context sequence is an effective way to break the linear relationship between the context length and the KV-cache Del Corro et al. (2023); Zhang et al. (2024); Anagnostidis et al. (2023); Sukhbaatar et al. (2019); Rae & Razavi (2020). The general intention of these algorithms is to find out the unimportant tokens in the sequence and drop the KV-cache of these tokens. For example, Sliding Window Attention Beltagy et al. (2020) only caches a certain number of the most recent tokens and drops the rest. StreamingLLM Xiao et al. (2023) found that in addition to the recent tokens, tokens at the beginning of the sequence are also crucial to the output. Heavy-Hitter Zhang et al. (2024) and Scissorhands Liu et al. (2024) rank the importance of tokens by comparing their attention scores. As mentioned above, these algorithms treat all the attention layers equally, and therefore, have a fixed KV-cache budget for each layer.

Optimizing the KV-cache on a batch basis mainly needs to manage the memory of different requests efficiently. For example, vLLM Kwon et al. (2023) allocates small chunks of memory in an on-demand way, instead of a fixed big block for each prompt, to reduce the memory fragmentation in a batch. RadixAttention Zheng et al. (2023) manages to share the KV-cache across requests in a batch when they have the same prefix in the prompt.

Last but not least, how to relax the linear relationship between the KV-cache and the layers remains largely unexplored compared with the other two dimensions. FastGen Ge et al. (2023) is a very recent work that found layers in different positions may have different optimal sequence-wise KV-cache eviction strategies. It then proposed an algorithm to choose the best eviction strategy from 1) Locality strategy (like Sliding Window), 2) Special Tokens strategy (like StreamingLLM), 3) Local and Frequency strategy (like Scissorhands, H2O) and so on for each attention head during the inference. However, for all the attention heads that have been assigned the same eviction strategy, FastGen simply gives them a unified pre-defined cache budget, like 30% of the sequence length. Therefore, how to adaptively allocate the KV-cache budget to layers with the same sequence-wise token eviction strategy is still largely unclear.

## 3 OBSERVATIONS

Inspired by some previous works that managed to optimize the LLM inference from a layer-wise perspective Del Corro et al. (2023); Ge et al. (2023), we make a hypothesis that attention layers in different positions play distinct roles in terms of importance, and therefore, should have different optimal KV-cache budgets. To better understand the layer-wise contribution to the output embedding, we employ cosine similarity as the metric to quantify the importance of each layer during inference. Specifically, in each attention layer, we track the hidden states of each input embedding at two spots, i.e., the embedding before and after the self-attention block. Denote the hidden state before the self-attention by $\mathbf{A}$, and the hidden state after the self-attention by $\mathbf{B}$. The cosine similarity between two embeddings $\mathbf{A}$ and $\mathbf{B}$ is calculated as follows:

$$\text{CosineSimilarity}(\mathbf{A}, \mathbf{B}) = \frac{\mathbf{A} \cdot \mathbf{B}}{\|\mathbf{A}\| \cdot \|\mathbf{B}\|} = \frac{\sum\limits_{i=1}^{n} A_i \cdot B_i}{\sqrt{\sum\limits_{i=1}^{n} A_i^2} \cdot \sqrt{\sum\limits_{i=1}^{n} B_i^2}} \tag{5}$$

where $\mathbf{A} = (A_1, A_2, \ldots, A_n)$ and $\mathbf{B} = (B_1, B_2, \ldots, B_n)$ are the vectors, and $n$ is the dimensionality.

For any given input embedding, we could get a cosine similarity of each attention layer that roughly quantifies how much the embedding changes after the self-attention computing of this layer. Then by comparing the cosine similarity among layers, we could find a way to rank attention layers by their importance. Following this line of thought, we choose 4 representative LLMs, i.e., Mistral-7B Jiang et al. (2023), Llama2-7B-32K Touvron et al. (2023), Llama2-70B Touvron et al. (2023), Falcon-7B Almazrouei et al. (2023), to conduct the experiments. Each model is fed by 200 prompts and we track the cosine similarity of each token in each layer. Figure 2 shows the results after averaging over prompts. Each row of the heatmap demonstrates how an input embedding at the given position evolves through all the layers. From the brightness of the figures, we can get some insights as follows: 1) In general, the first half of layers (in a darker color) contributes more to the output embeddings than the second

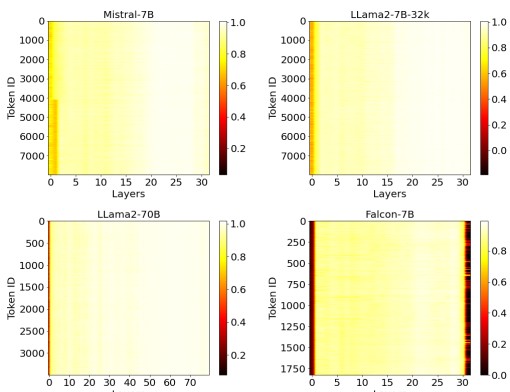

Figure 2: Visualization of cosine similarity before and after the self-attention calculation of attention each layer. The layers with higher cosine similarity, represented by lighter colors, exert a relatively lower impact on the input vectors.

half of layers (in a lighter color) does; 2) The first and last few layers tend to be more critical than other layers, depending on the specific models and tasks; 3) The cosine similarity can effectively depict the layer-wise importance, as the trend it reflects aligns with the previous studies, like Early-exiting Del Corro et al. (2023) and FastGen Ge et al. (2023).

This observation gives us a simple yet effective metric to design a new algorithm that is able to optimize the KV-cache from not only the context's dimension, but also the layer's dimension.

## 4 ALGORITHM

In this section, we describe the SQUEEZEATTENTION algorithm inspired by our observations from the last chapter. The most distinct feature of the proposed algorithm is that it considers tokens in the KV-cache as a 2D matrix with one dimension of sequence and another dimension of layer, and both dimensions are going to be optimized jointly.

## 4.1 SQUEEZEATTENTION

In the sequence's dimension, there are various cache eviction policies that we could directly combine with, like Least Recently Used methods (Sliding Window, StreamingLLM), Least Frequently Used methods (Scissorhands, H2O), and so on. We denote a policy that compresses the KV-cache in sequence's dimension by $\boldsymbol{C}_{seq}$, and its cache budget by $b_{init}$. Note that all the layers have the same cache budget by default, just like the assumptions made by all these sequence-wise KV-cache compression algorithms.

In the layer's dimension, SQUEEZEATTENTION firstly tracks the layer importance with the given prompt in the prefilling phase by collecting the cosine similarities of each layer whenever the self-attention is conducted. At the end of prefilling, each layer ends up with a set of cosine similarities, each of which corresponds to a token that has flowed through this layer. Then we use the averaged value over prompt tokens to represent the layer-wise importance of this layer regarding this task. By clustering the layers into groups based on the layer-wise importance with KMeans, we reallocate the $b_{init}$ for each layer in a way that more budgets are assigned to the more "important" layer groups. Since layers have different cache budgets, in the decoding phase, $\boldsymbol{C}_{seq}$ works separately with each layer's very own budgets. The detailed process is described in Algorithm 1.

---

**Algorithm 1** SQUEEZEATTENTION

**Require:** $prompt$: input sequence; $\boldsymbol{C}_{seq}$: a KV-cache compressor in sequence dimension; $b_{init}$: the initial cache budget of each layer; $n_{layer}$: number of attention layers; $p$: hyperparameter ($0 < p < 1$); $K^{(i)}$: KV-cache of the $i$-th layer;

1: Feed the $prompt$ into the model for **prefilling**, calculate $cos\_sim_j^{(i)}$ of the $j$-th token in the $i$-th layer by Equation 5;
2: **for** $i \leftarrow 1$ to $n_{layer}$ **do**
3:      $cos\_sim^{(i)} = \frac{\sum_{j=1}^{\texttt{len}(prompt)} cos\_sim_j^{(i)}}{\texttt{len}(prompt)}$
4: **end for**
5: $G_1, G_2, G_3 \leftarrow \texttt{KMeans}(cos\_sim^{(i)})$            ▷ Cluster layers into 3 groups by $cos\_sim^{(i)}(1 \leqslant i \leqslant n_{layer})$, where $G_3$ has the biggest $cos\_sim$ on average
6: **for** $i \leftarrow 1$ to $n_{layer}$ **do**
7:      **if** $i \in G_3$ **then**
8:          $b^{(i)} = b_{init} \times p$
9:      **else**
10:          $b^{(i)} = \frac{n_{layer} \times b_{init} - \texttt{len}(G_3) \times b_{init} \times p}{\texttt{len}(G_1) + \texttt{len}(G_2)}$
11:      **end if**
12:      $K^{(i)} = \texttt{KV}(prompt)$            ▷ the prompt's KV is cached after prefilling
13: **end for**
14: **for** $o \leftarrow 1$ to $\texttt{len}(output)$ **do**            ▷ **Decoding** output tokens one by one
15:      **for** $i \leftarrow 1$ to $n_{layer}$ **do**
16:          **if** $\texttt{len}(K^{(i)}) > b^{(i)}$ **then**
17:             $K^{(i)} = \boldsymbol{C}_{seq}(K^{(i)}, b^{(i)})$    ▷ Compress the KV-cache of this layer by its own budget
18:          **end if**
19:          Finish the Self-attention based on the compressed $K^{(i)}$.
20:      **end for**
21: **end for**

---

## 4.2 DISCUSSIONS

### 4.2.1 HOW TO DECIDE THE VALUE OF $p$

SQUEEZEATTENTION involves a hyperparameter $p$ to control the percentage of initial budgets that could be removed from the "unimportant" layers. The smaller the $p$ is, the more budgets will be re-assigned. In experiments, we found 0.3-0.4 is a reasonable choice range in most cases. To precisely understand the impact of $p$, we have conducted extra experiments to demonstrate how the model accuracy changs with the value of $p$, please refer to A.2 for more details.

### 4.2.2 WHY CLUSTERING INTO THREE GROUPS?

Based on the observations of 7 models we have tried, we found they all have a typical pattern (3 groups) with respect to the layer importance. Specifically, Group 1 consists of a few special layers (always the first and last few layers) which can be seen as an analogy of special tokens that should never be evicted. Then Group 2 and Group 3 do not have a fixed borderline with each other, but we found that Group 3 makes obviously less impact on the embeddings, which can be seen as an analogy of "frequent" and "unfrequent" tokens in the sequence-wise methods. Therefore, our 3-group policy could be rephrased as: "we firstly identify and prioritize the special layers (Group 1), then classify the rest layers into two groups: important (Group 2) and unimportant (Group 3), then reallocate the cache budget based on the clustering. Even if we cluster the layers into more than 3 groups, we just break down Group 2 and Group 3 into many small groups, but they eventually need to be reduced into two classes again, that is, either reducing the budget or increasing the budget. To preserve the model accuracy, we only reduce cache budgets from Group 3, which accounts for around 50% to 70% of total layers.

Table 1: Datasets used in our experiments.

| Task | Task Type | Eval metric | Avg len | Language | Sample |
|------|-----------|-------------|---------|----------|--------|
| CNN/ Daily Mail | Summarization (3 sentence) | Rouge-2 | 2,000 | EN | 1,000 |
| XSUM | Summarization (1 sentence) | Rouge-2 | 2,000 | EN | 1,000 |
| SAMSUM | Few shot | Rouge-L | 6,258 | EN | 200 |
| NarrativeQA | Single-doc QA | F1 | 18,409 | EN | 200 |
| TriviaQA | Few shot | F1 | 8,209 | EN | 200 |

## 5 EXPERIMENTS

### 5.1 EXPERIMENT SETUP

**LLM Models.** We choose 7 representative LLMs, with model sizes ranging from 6.7B to 70B and context lengths ranging from 2K to 32K, to evaluate the proposed algorithm: GPT-NeoX-20B, OPT-6.7B, Falcon-7B, Mistral-7B, Mixtral-8×7B, LLama2-7B-32k, LLama2-70B.

**Datasets.** We conduct experiments on 5 datasets: CNN/Daily Mail, XSUM, TriviaQA, SAMSUM, and NarrativeQA. TriviaQA, SAMSUM, and NarrativeQA originate from LongBench Bai et al. (2023), where the data length typically exceeds 8k. CNN/Daily Mail and XSUM have an average length of about 2k. Detailed information about datasets can be found in Table 1.

**Baselines.** 3 sequence-wise sparsification algorithms are chosen as the baselines to integrate into SQUEEZEATTENTION and we assign each algorithm to the model that works beat, which we call the beat baseline algorithms:

- Heavy-Hitter (H2O) Zhang et al. (2024): Identify the important tokens within the sequence by comparing the accumulated attention score of each token.

- Sliding Window Attention Beltagy et al. (2020): A "Local" strategy that only caches the most recent tokens' KV embeddings. This method works especially well with Mistral and Mixtral.

- StreamingLLM Xiao et al. (2023): In addition to the most recent tokens, StreamingLLM always caches the first $n$ tokens in the sequence, as they are identified as "sink tokens". We take $n = 4$ as recommended by the paper.

**Hardwares.** We conduct all the experiments on the AWS platform (p4d.24xlarge) with 8 Nvidia A100-40GB GPUs, interconnected by the NVLinks (600 GB/s GPU peer-to-peer bandwidth).

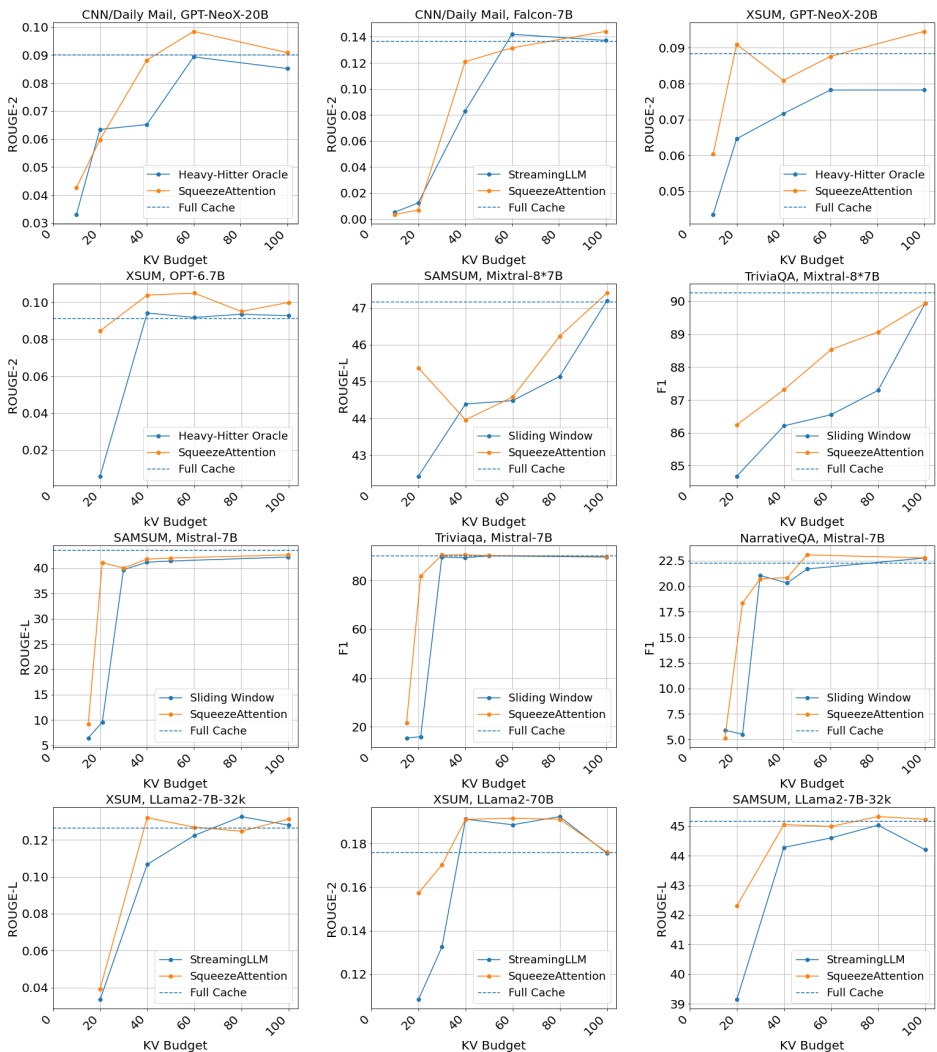

Figure 3: Performance of SQUEEZEATTENTION, best sequence-wise compression baselines, and Full Cache under different cache budgets.

## 5.2   END-TO-END RESULT

Figure 3 demonstrates the comparison results of SQUEEZEATTENTION and other 3 baseline algorithms over all 7 models and 5 datasets. Full Cache (dashed line) means all tokens' KV embeddings are fully cached during the inference, therefore, it represents a relatively good model accuracy. The blue and orange curves in each subfigure illustrate how the model accuracy changes with the KV-cache budgets ranging from 10% to 100% of the total sequence length. Note that applying different sequence-wise compression algorithms to different tasks would lead to quite different model accuracies. Therefore, for each task, we choose the best sequence-wise compression algorithm to represent the best case, and then apply SQUEEZEATTENTION on top of the best case. As shown in the figure, SQUEEZEATTENTION consistently manages to improve the model accuracies under various KV-cache budgets by reallocating the cache budgets among layers. In other words, SQUEEZEAT-TENTION can achieve similar inference accuracies with much less KV-cache in total.

## 5.3   MEMORY CONSUMPTION

Now we evaluate the memory consumption of the proposed algorithm. We choose three settings to compare how much GPU memory it needs to run the inference without degradation of model

Table 2: Comparisons of the required KV-cache budget to achieve the best accuracy. Three models are selected to represent the small (7B), medium (20B), and large models (70B). For each task, we choose the best existing sequence-wise sparsification algorithm as the baseline.

| Model | Size | Dataset | Best Baseline | Performance / Used KV Budget | | |
|-------|------|---------|---------------|------------------------------|--|--|
| | | | | Full Cache | w/ SQUEEZEATTENTION | w/o SQUEEZEATTENTION |
| Mistral | 7B | SAMSUM | Sliding Window | 43.53 / 100% | 41.05 / **20%** | 40 / 30% |
| GPT-NeoX | 20B | XSUM | Heavy-Hitter Oracle | 0.09 / 100% | 0.09 / **20%** | 0.08 / 60% |
| LLama2 | 70B | XSUM | StreamingLLM | 0.18 / 100% | 0.17 / **30%** | 0.19 / 40% |

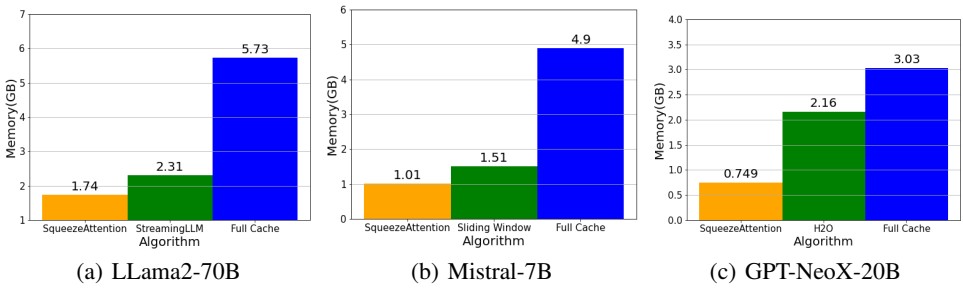

(a) LLama2-70B      (b) Mistral-7B      (c) GPT-NeoX-20B

Figure 4: Comparisons of per-token decoding memory usage (exclude model parameters) among the Full cache, SQUEEZEATTENTION, and best baselines in order to achieve the same accuracy, as shown in Table 2.

accuracy. We select Mistral-7B (Sliding Window), GPT-NeoX-20B (Heavy-Hitter), and LLama2-70B (StreamingLLM) to cover all three baseline algorithms and models in small, middle, and large sizes. We utilize multi-GPU inference if the model and KV-cache do not fit into a single GPU.

Table 2 demonstrates that in all three settings, by achieving the same model accuracy, SQUEEZEAT-TENTION consumes the least amount of KV-cache budgets compared with the algorithms that only compress from the sequence's dimension. In some cases, it only takes one-third of the cache budget of H2O algorithm. Subsequently, we employ **PYTORCH PROFILER** to evaluate the diminished memory usage of generating one token during the inference (w/o memory usage of model weights). Figure 4 shows that SQUEEZEATTENTION can save 70% to 80% of memory usage per token compared with Full Cache method, and 25% to 66% of memory usage compared with baseline algorithms.

## 5.4 THROUGHPUT OF TOKEN GENERATION

Since SQUEEZEATTENTION manages to save the memory cost of inference, as shown in the previous sections, we want to explore how these memory reductions can be interpreted into improvements of token throughput. We choose 2 models, i.e., Mistral-7B and Llama-70B, to represent models in small and large sizes. With a fixed content length, we increase the batch size from 1 to 224 for Mistral-7B, and 1 to 64 for Llama2. For each task, we select the best baseline algorithm, that is, Sliding Window for Mistral-7B and StreamingLLM for Llama2-70B.

Table 3 shows the token throughput on 8 A100-40GB GPUs. With the same batch size, SQUEEZEATTENTION can enhance throughput by up to 2.2× for Mistral-7B and 1.4× for Llama2-70B compared to the Full Cache. Besides, SQUEEZEATTENTION also enables batch sizes up to 224 and 64 for two models, which would cause out-of-memory for the Full Cache method. The throughput comparison between best baseline and SQUEEZEATTENTION is reported in A.4.

## 5.5 OVERHEAD OF THE ALGORITHM

The computational overhead of SQUEEZEATTENTION comes from two operations: Cosine similarity and Kmeans. The execution of all these computations only happens during the prefilling phase.

Table 3: Generation throughput (token / s) on eight A100 GPUs of Mistral-7B and LLama2-70B with SQUEEZEATTENTION and Full Cache. To maintain the model accuracy, SQUEEZEATTENTION uses 20% of the cache budget for Mistral-7B and 30% of the cache budget for LLama2-70B. "OOM" means out-of-memory. More results can be found in A.4.

| Model | Size | prompt len + gen len | Algorithm | Batch size | | | | |
|-------|------|----------------------|-----------|------|------|------|------|------|
| | | | | 1 | 32 | 64 | 128 | 224 |
| Mistral | 7B | 512 + 1024 | SQUEEZEATTENTION | 20.5 | 496.5 | 682.7 | 824.4 | 892.5 |
| | | | Full Cache | 20.9 | 254.0 | 304.8 | OOM | OOM |

| Model | Size | prompt len + gen len | Algorithm | Batch size | | | | |
|-------|------|----------------------|-----------|------|------|------|------|------|
| | | | | 1 | 8 | 16 | 32 | 64 |
| LLama2 | 70B | 256 + 512 | SQUEEZEATTENTION | 5.2 | 37.2 | 71.2 | 116.2 | 170.7 |
| | | | Full Cache | 5.2 | 36.0 | 62.5 | 84.8 | OOM |

Therefore, the cost of SQUEEZEATTENTION is a one-time price, which is much more cost-efficient than those algorithms (like H2O) that require extra calculations in each iteration of token generation.

We profiled the entire prefilling phase of Mistral-7B with/without SQUEEZEATTENTION to compare the wall-clock time. The prompt length is up to 8k tokens. As shown in Table 4, the SQUEEZEATTENTION only caused 6.3% increasement of **prefilling time**. Note that if we take the decoding time into consideration, this ratio is going to be much smaller, depending on the actual number of tokens generated. Therefore, the overhead of SQUEEZEATTENTION is basically neglectable. The more detailed overhead of SQUEEZEATTENTION can be found in A.1.

Table 4: Overhead of SQUEEZEATTENTION (Prefilling Time in seconds on one Nvidia A100-40GB)

| Model | w/o SQUEEZEATTENTION | w/ SQUEEZEATTENTION | Overhead Ratio |
|-------|----------------------|---------------------|----------------|
| Mistral-7B | 0.636 | 0.676 | 6.3% |

## 5.6 LIMITATIONS AND BROADER IMPACTS

SQUEEZEATTENTION works jointly with a sequence-wise sparsification policy, so the assumption we make is that for a given model and dataset, there exists a sequence-wise KV-cache eviction policy that won't hurt the model accuracy under a certain tolerance. However, the generalizability of these sequence-wise algorithms is still an active research topic. If this assumption cannot be met, SQUEEZEATTENTION might not work as expected.

The positive social impacts include reducing the energy cost and carbon emissions of LLM inference. Besides, it can optimize the user experience of LLM applications. Since the proposed algorithm could accelerate the LLM inference, the potential negative impact is that it might exaggerate the improper usage of LLM, like generating harmful information.

## 6 CONCLUSION

In this paper, we propose a 2D KV-cache compression algorithm called SQUEEZEATTENTION. By tracking the cosine similarity of each attention layer, we found that layers in different positions have distinct degrees of importance regarding the output embedding. Inspired by this observation, SQUEEZEATTENTION reallocates the KV-cache budgets over attention layers to further reduce the memory cost of inference. Experiments over a wide range of models and tasks show that SQUEEZEATTENTION can achieve better model accuracies with lower memory consumption compared with state-of-the-art algorithms that only compress KV-cache in a sequence-wise way.

ACKNOWLEDGEMENT

This work is supported by National Natural Science Foundation of China (U23B2048, U22B2037), the Fund of Kunpeng and Ascend Center of Excellence (Peking University), and High-performance Computing Platform of Peking University.

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

# A APPENDIX

## A.1 DETAILED OVERHEAD OF SQUEEZEATTENTION

In this section, we further evaluate the overhead introduced by SQUEEZEATTENTION, we broke down the time taken for two primary operations: cosine similarity computation and K-means clustering. The experiment setup follows the same setup as 5.5. This experiment was conducted using a single Nvidia A100-40GB GPU with prompt lengths of up to 8k tokens.

Table 5: detailed onverhead

| Cosine similarity | K-means | Total time |
|---|---|---|
| 0.00068s | 0.001s | 0.02276s |

The operation of computing cosine similarity involves calculating between two arrays of size 8000×4096, repeated 32 times (since Mistral has 32 layers). Additionally, K-means clusters 32 numbers into 3 classes. Therefore, the total time consumption can be calculated as 0.00068×32+0.001=0.02276 seconds. It is noteworthy that this overhead is incurred only once, regardless of the number of tokens processed.

## A.2 THE FUNCTION OF $p$

The hyperparameter $p$ is crucial in the SqueezeAttention mechanism, as it directly determines the final allocation of the KV cache. To illustrate this with a concrete example: suppose we have a model with 32 layers, where 18 layers are deemed important and the remaining 14 are considered less important. Each layer initially has a budget of 1000 tokens. If we set $p$ to 0.3, we will take 70% of the budget from the less important layers and redistribute it equally among the important layers.

Specifically, the budget for the less important layers will be reduced to 1000×0.3=300. Meanwhile, the budget for the important layers will be calculated as follows: (1000×18+1000×0.7×14)/18=1544. In this way, the total budget remains unchanged.

To evaluate the sensitivity of the hyperparameter $p$, we tested the Mistral-7B model on the Samsum dataset with $p$ values ranging from 0.1 to 1.0. The total KV budget was set to 20% of the prompt length. The results are presented in the table below:

Table 6: The precision changing with $p$

| ROUGE-L | 15.7 | 34.01 | 37.26 | **37.69** | 27.41 | 26.29 | 11.11 | 10.48 | 8.72 | 9.07 |
|---|---|---|---|---|---|---|---|---|---|---|
| p | 0.1 | 0.2 | 0.3 | **0.4** | 0.5 | 0.6 | 0.7 | 0.8 | 0.9 | 1.0 |

100% means we do not alter the structure of the model's KV cache. As $p$ decreases, more KV budget will be transferred to other layers, which can yield better results up to a certain point. When $p$ decreases to 10% or even lower, the performance decreases as the less important layers' KV budget becomes severely inadequate. We can clearly observe an accuracy improvement when adjusting $p$ while keeping the overall budget unchanged. This demonstrates the impact of $p$ on model performance, highlighting its importance in optimizing KV budget allocation.

## A.3 LAYER IMPORTANCE ACROSS DIFFERENT TASK

We conducted an extra experiment using two models and various datasets to determine whether the importance of different layers is an intrinsic property of the model.

Table 7: Mistral-7B

| Dataset | Samsum | TriviaQA | LCC |
|---|---|---|---|
| Important | 17 | 18 | 19 |
| Unimportant | 15 | 14 | 13 |

Table 8: LLama2-70B

| Dataset | Xsum | Samsum | LCC |
|---|---|---|---|
| Important | 17 | 21 | 18 |
| Unimportant | 63 | 59 | 62 |

Tabel7 displays the distribution of important layers for the Mistral model across three different datasets: Samsum (Few shot), TriviaQA (Single-document QA), and LCC (Code, Python/C#/Java) and table 8 shows the distribution of important layers for the LLama2-70B model across three different datasets: Xsum (Summarization), Samsum (Few shot), and LCC (Code, Python/C#/Java).

From these tables, we can observe that there is a rough pattern regarding the layer's group with task-specific fluctuations. We believe there exist some task-sensitive layers that may be classified into different groups with different tasks. Similarly, there are also some layers that are always important / unimportant. A detailed analysis of this phenomenon could be an interesting extension of this work. However, we would still recommend an adaptive way since it can precisely capture the importance of layers.

## A.4 THROUGHPUT COMPARISON BETWEEN BEST BASELINE ALGORITHMS AND SQUEEZEATTENTION

Table 9: Generation throughput (token / s) on eight A100 GPUs of Mistral-7B and LLama2-7B with SQUEEZEATTENTION and best baseline algorithms. To maintain the model accuracy, SQUEEZEATTENTION uses 20% of the cache budget for Mistral-7B and 40% of the cache budget for LLama2-7B while Sliding Window uses 30% of the cache budget for Mistral-7B and StreamingLLM use 60% of the cache budget for LLama2-7B. "OOM" means out-of-memory.

| Model | Size | prompt len + gen len | Algorithm | Batch size | | | | |
|---|---|---|---|---|---|---|---|---|
| | | | | 1 | 32 | 64 | 128 | 224 |
| Mistral | 7B | 512 + 1024 | SQUEEZEATTENTION | 20.5 | 496.5 | 682.7 | 824.4 | 892.5 |
| | | | Sliding Window | 20.6 | 404.5 | 512.2 | 587.8 | OOM |

| Model | Size | prompt len + gen len | Algorithm | Batch size | | | | |
|---|---|---|---|---|---|---|---|---|
| | | | | 1 | 32 | 64 | 128 | 256 |
| LLama2 | 7B | 512 + 1024 | SQUEEZEATTENTION | 20.0 | 143.0 | 150.4 | 144.9 | OOM |
| | | | StreamLLM | 20.4 | 113.7 | 102.4 | OOM | OOM |

We also conducted experiments to compare throughput of SQUEEZEATTENTION with best baseline under a set of batch sizes. Both experiments used an input length of 512 and an output length of 1024. We choose the compression hyperparameters for each algorithm such that they could all achieve the best mode accuracy. The result show that our algorithm can obviously increase the throughput compared with those SOTA algorithms that only compress KV-cache from the sequence's dimension.

