# OpenReview forum: "SqueezeAttention: 2D Management of KV-Cache in LLM Inference via Layer-wise Optimal Budget"
_ICLR.cc/2025/Conference — ICLR 2025 Poster_

### Official Review · Reviewer_Pb9Y · 2024-10-29

**Soundness:** 4
**Presentation:** 4
**Contribution:** 3
**Rating:** 8
**Confidence:** 4

**Summary:**

The paper proposes SqueezeAttention, a novel 2D Key-Value (KV) cache management algorithm designed to optimize memory usage and processing efficiency during Large Language Model (LLM) inference. The motivation behind this work is that existing KV-cache compression strategies handle all attention layers equally, which is suboptimal. Instead, SqueezeAttention dynamically allocates the KV-cache budget based on each layer's importance, determined by the cosine similarity of embeddings before and after each self-attention layer. By combining sequence-wise and layer-wise cache optimization, SqueezeAttention provides substantial memory savings (30%-70%) and throughput improvements (up to 2.2×) across various LLM models, including Mistral-7B, Falcon-7B, and Llama2-70B. The experimental results show significant performance gains in memory efficiency and token generation speed.

**Strengths:**

Novel Layer-Wise Approach: This paper introduces a layer-wise approach to KV-cache optimization, differentiating it from existing sequence-based compression methods. This work fills a gap in current LLM efficiency research.
Significant Performance Improvement: The proposed method improves memory consumption and throughput by reallocating cache budgets based on layer importance.
Robust Experimental Validation: The authors test their approach on multiple models (ranging from 7B to 70B parameters) and datasets, demonstrating its generalizability and efficiency.
Compatibility with Other Methods: SqueezeAttention integrates smoothly with various sequence-wise compression techniques, enhancing its versatility.
Energy Efficiency: The memory and throughput improvements have practical implications, potentially reducing the environmental impact of LLM deployment.

**Weaknesses:**

Dependency on Sequence-Wise Algorithms: The effectiveness of SqueezeAttention relies on combining it with existing sequence-wise compression methods, which limits its standalone applicability.
Potential Task-Specific Tuning: Although the layer importance measurement is automated, there may be task-specific variations, suggesting possible limitations in generalizing to unseen tasks without fine-tuning.
Limited Analysis of Computational Overheads: Although the paper claims that SqueezeAttention adds a negligible overhead, more analysis on computation costs, particularly for real-time applications, would strengthen the results.
Fixed Group Clustering: The choice of clustering layers into three fixed groups may oversimplify the optimization for some models or tasks where layer importance does not align neatly with this structure.
Risk of Reduced Accuracy: The method risks performance degradation for certain parameter values by under-allocating cache to less "important" layers, which might be essential for specific tasks or models.
In conclusion, the paper presents a promising contribution to LLM inference optimization with its innovative, adaptive KV-cache management strategy. However, further exploration into standalone performance and task-specific tuning would enhance the robustness of SqueezeAttention.

**Questions:**

See the discussion of weaknesses and kindly address them.

---

> ### Author Response · Authors · 2024-11-17
> **response to reviewer Pb9Y (part 1)**
>
> **Weaknesses：**
>
> 1. Dependency on Sequence-Wise Algorithms: The effectiveness of SqueezeAttention relies on combining it with existing sequence-wise compression methods, which limits its standalone applicability.
>
> **Answer:**
>
> We designed SqueezeAttention to focus solely on determining the KV cache budget for layers, with the goal of making it a versatile tool that can be seamlessly integrated with the existing landscape of sequence-wise compression methods, because they are two orthogonal dimensions of this problem. However, once SqueezeAttention has adaptively decided the cache budget for each layer (this process is standalone), the sequence-wise eviction could be as simple as Sliding Window (least recently used cache). Such a simple token eviction strategy can work quite well with SqueezeAttention in many cases.
>
> 2. Potential Task-Specific Tuning: Although the layer importance measurement is automated, there may be task-specific variations, suggesting possible limitations in generalizing to unseen tasks without fine-tuning
>
> **Answer:**
>
> Task-specific tuning can indeed enhance accuracy for unseen tasks, but it would require additional research and development to implement effectively within SqueezeAttention. We see this as a promising future direction, as refining the method to adapt to diverse tasks could significantly improve its generalization capabilities.
>
> 3. Limited Analysis of Computational Overheads: Although the paper claims that SqueezeAttention adds a negligible overhead, more analysis on computation costs, particularly for real-time applications, would strengthen the results.
>
> **Answer:**
>
> To assess the computational overhead introduced by SqueezeAttention, we measured the time taken to generate the first token with and without SqueezeAttention enabled.
>
> | Model | With SqueezeAttention | Without SqueezeAttention |
> | --- | --- | --- |
> | Mistral-7B (Sliding Window) | 0.636s | 0.676s |
>
> This experiment was conducted on a single Nvidia A100-40GB GPU with prompt lengths of up to 8k tokens. As shown above, the difference in time between the two scenarios is minimal.
>
> Additionally, we analyzed the specific overhead introduced by SqueezeAttention, which is primarily due to two operations: cosine similarity and K-means clustering.
>
> | Operation | Time (seconds) |
> | --- | --- |
> | Cosine Similarity | 0.00068s |
> | K-means Clustering | 0.001s |
> | Total Overhead | 0.02276s |
>
> The cosine similarity computation involves two arrays of size 8000x4096, repeated 32 times (for each layer in the Mistral model), and K-means clustering is used to group 32 numbers into 3 clusters. The total overhead is therefore calculated as 0.00068×32+0.001=0.02276 seconds. This additional overhead is incurred only once, regardless of the number of tokens.
>
> 4. Fixed Group Clustering: The choice of clustering layers into three fixed groups may oversimplify the optimization for some models or tasks where layer importance does not align neatly with this structure.
>
> **Answer:**
>
> This is a great question. Based on the observations of 7 models we have tried, we found they all have a typical pattern (3 groups) with respect to the layer importance. Specifically, Group 1 consists of a few special layers (**always the first and last few layers**) which can be seen as an analogy of special tokens that should never be evicted (like the "sink token" found in StreamingLLM). The cosine similarity values of Group 1 tend to differ significantly from those of other layers. Then Group2 and Group3 do not have a fixed borderline with each other, but we can see that Group3 makes obviously less impact on the embeddings then Group2 does, which can be seen as an analogy of "frequent" and "unfrequent" tokens in the sequence-wise methods. **Therefore, our policy could be rephrased as:** "firstly identify the special layers (**Group1**), then classify the rest layers into two groups: important (**Group2**) and unimportant (**Group3**), then reallocate the cache budget based on the clustering result. Even if we cluster the layers into more than 3 groups, we are just breaking down Group2 and Group3 into small sub-groups, but  eventually they need to be reduced into two classes again, that is, either reducing the budget or increasing the budget.

---

> > ### Author Response · Authors · 2024-11-17
> > **response to reviewer Pb9Y (part 2)**
> >
> > 5. Risk of Reduced Accuracy: The method risks performance degradation for certain parameter values by under-allocating cache to less "important" layers, which might be essential for specific tasks or models.
> >
> > **Answer:**
> >
> > We acknowledge that there is a potential risk of performance degradation if cache allocation to certain layers is insufficient, as some "less important" layers may be crucial for specific tasks or models. While our extensive testing across a range of tasks and models has demonstrated that SqueezeAttention can effectively handle various scenarios, there may still be cases where it does not perform optimally. Nonetheless, we believe that SqueezeAttention has shown strong adaptability across diverse tasks and holds promise for further refinement and improvement in future research.

---

### Official Review · Reviewer_fGAX · 2024-11-03

**Soundness:** 3
**Presentation:** 2
**Contribution:** 1
**Rating:** 5
**Confidence:** 3

**Summary:**

This paper identifies the importance of different attention layers, and proposes a layer-wise strategy named as SqueezeAttention to allocate different KV cache size for each layer. However, the proposed method still have several significant issues.

**Strengths:**

The observations of comparing the inputs and outputs of attention modules are good.

**Weaknesses:**

1.	The proposed method is designed only for the prefilling stage and does not allow for dynamic adjustment of the KV cache size during the decoding stage. To improve applicability, it would be helpful if the authors discussed potential ways to extend the method to the decoding stage, or provided a rationale explaining why it may not be feasible in that context.
2.	The reduced KV cache size is controlled by the hyperparameter \( p \), with values in the range of 0.3-0.4 based on a single model and task. This approach lacks generality. To improve robustness, the authors could conduct experiments across multiple models and tasks to determine if this \( p \) value range holds more broadly. Alternatively, they could propose a method for automatically selecting \( p \) to adapt to different scenarios.
3.	The method uses a fixed number of clusters, specifically 3, which may limit its generalizability. To strengthen the justification for this choice, the authors could either provide a rationale for using 3 clusters or experiment with different numbers of clusters to determine the optimal setting across various scenarios.
4.	The experiments appear incomplete. While Figure 3 includes four baselines, such as the full KV cache, each experiment only presents one baseline alongside the proposed method for comparison. Including all baselines in each experiment would allow for a more comprehensive evaluation. If certain baselines were omitted, the authors should explain why.

**Questions:**

It is unclear why the authors use ROUGE-2 for CNN/Daily Mail and XSUM, but ROUGE-L for SAMSUM. ROUGE-L is generally considered a more accurate metric for summarization tasks and could be applied consistently across all datasets. The authors could either evaluate all datasets with ROUGE-L for consistency or provide a rationale for choosing different metrics for each dataset.

---

> ### Author Response · Authors · 2024-11-13
>
> Thank you for taking the time to review my paper on **SqueezeAttention**. However, I noticed that your comments seem to be based on a different paper, specifically one that proposes **BBOPlace-Bench**, a benchmark for evaluating and developing a black box optimization for chip placement for EDA.
>
> I believe there may have been a misunderstanding or mix-up in the review process. Could you please take a moment to review my paper again and provide comments that are relevant to the content and research presented?

---

> ### Author Response · Authors · 2024-11-23
> **response to reviewer fGAX**
>
> Dear Reviewer,
>
> Thank you again for your valuable feedback. This is a kind reminder that we have conducted additional experiments to address the concerns you raised and have further clarified the different metric used in our experiment.
>
> We would greatly appreciate it if you could kindly reconsider the assessment. Please feel free to reach out if you have any further questions or require additional clarifications.
>
> Additionally, it seems that our contribution score is still being evaluated based on the scores of BBOPlace-Bench. Could you kindly confirm if this has been updated in the latest assessment?
>
> Thank you for your time and support!

---

> ### Author Response · Authors · 2024-12-04
> **Thanks for your Reviews**
>
> Dear fGAX Reviewer,
>
> Thanks again for your reviews and suggestions. Since the discussion phase is going to end soon, we'd like to summarize our rebuttal content for your consideration.
> 1. Explanation about why the KV cache budget can be decided by prefilling only.
> 2. Additional thorough experiments regarding the selection of the hyper-parameter p introduced by our algorithm.
> 3. Explanation about why the number of clusters should be 3 without hurting the generalizability.
> 4. The design principle behind the Figure 3 that we only choose the best baseline algorithm to compare with for each task, because not every baseline algorithm can actually work for each task.
> 5. Explanation about the choice of different metrics for different tasks.
>
> Besides, since there was a mixing-up of reviews, we'd like to kindly ask, if those sub-scores of Soundness, Presentation, and Contribution have also been corrected?
> Please let us know if there are any further questions or suggestions. Thanks!

---

### Official Review · Reviewer_2BQ8 · 2024-11-06

**Soundness:** 2
**Presentation:** 3
**Contribution:** 2
**Rating:** 3
**Confidence:** 5

**Summary:**

This work proposed a layer-wise KV cache compression method that reduce the overhead during decoding stage of LLM inference. The proposed squeezeattention use cosine similarity of embeddings before and after attention block to identify the redundancy of kv cache with respect to specific layer. Then more redundant layers will then be assigned with smaller kv cache budget. For each layer, squeezeattention based on previous methods to remove redundant kv pairs, such as H2O, SteamingLLM and Sliding windows.

**Strengths:**

- The proposed methods is evaluated with multiple LLMs on various downstream tasks, demonstrates non-trivial improvements against previous baselines.

- The manscript is clearly organized with several illustration figures and equations. It's easy to understand the main method of this work.

- Both perfomance comparison and end-to-end memory/thoughput comparison are reported.

**Weaknesses:**

- The main observation that the cosine similarity of embeddings changes across layers while the first and last layers tend to have more diverse embeddigns, is not very new. Several works have showed similar results[1-3].

- It would be helpful to consider more recent kv cache compression methods, like SnapKV, PyramidKV, KIVI, etc. As the layer-wise strategy seems can be used in either KV cache pruning/quantization/low-rank decomposition methods, etc.

- In Table 3, it's a little bit unfair to compare the thoughput only with the full cache, since the KV cache evicted method is not the contribution of this work while the part of the thoughput improvements is achieved by the kv eviction, rather than the layer-wise strategy.


[1] https://arxiv.org/abs/2312.17276

[2] https://proceedings.neurips.cc/paper_files/paper/2023/file/fde1a69a5b6e554b2f1f727197d2651d-Paper-Conference.pdf

[3] https://arxiv.org/pdf/2202.08625

**Questions:**

- Do G1,G2,G3 changes frequently across different samples? otherwise we can assign the layer-wise budget through a offline process.

---

> ### Author Response · Authors · 2024-11-17
> **response to reviewer2BQ8 (part 1)**
>
> Dear reviewer, thank you very much for your comments and professional advice. Based on your suggestion and request. We added additional experiments to clarify some of the ambiguous or overlooked aspects and further explain SqueezeAttention and other similar methods. We would like to provide the details as follows:
>
> **Weaknesses:**
>
> 1. The main observation that the cosine similarity of embeddings changes across layers while the first and last layers tend to have more diverse embeddigns, is not very new. Several works have showed similar results.
>
> **Answer:**
>
> Thank you for mentioning these valuable related works. We totally agree that the pattern of token representations over attention layers (measured by cosine similarity) have been studied in previous works, since it’s such an intrinsic character of self-attention models. However, we’d like to kindly highlight our novelty in three aspects:
>
> - Different motivation and problem. People dive into the embeddings across layers for quite different motivations. Some aim to mitigate the over-smoothing problem, as cited by reviewer. Some aim to reduce the computation cost, like early-exiting. But we find out that the massive memory cost of inference could also be the beneficiary of this phenomenon, which indicates the novelty of our work.
> - Given this observation, it’s non-trivial to design a practical solution for improving inference efficiency. Inspired by previous works, our contribution is mainly about how to take advantage of this intrinsic to reduce the memory cost of KV cache, with careful consideration of the existing landscape of efficient inference algorithms. We manage to balance the generalisability and efficiency, ending up with an end-to-end solution.
> - We also extend the knowledge regarding how token embeddings evolve through layers with different models and tasks. Although the high-level pattern is that token embeddings tend to be more similar as layer goes deeper, we find that this monotonicity does not always hold. For example, the similarity may have a sudden decrease in some middle or deep layers for some models and tasks. This is crucial when designing the algorithm that intends to rank the potential priority of attention layers or heads.
>
> 2.  It would be helpful to consider more recent kv cache compression methods, like SnapKV, PyramidKV, KIVI, etc. As the layer-wise strategy seems can be used in either KV cache pruning/quantization/low-rank decomposition methods, etc.
>
> **Answer:**
>
> Thank you for highlighting recent KV cache compression methods like SnapKV, PyramidKV, and KIVI. We appreciate your insight that our layer-wise strategy can be applied to various approaches, including pruning, quantization, and low-rank decomposition. This flexibility indeed reflects the potential of our algorithm to enhance existing methods by serving as a complementary optimization strategy. I want to further explain the feasibility and benefit of combination of SqueezeAttention and these algorithms.

---

> > ### Author Response · Authors · 2024-11-17
> > **response to reviewer2BQ8 (part 2)**
> >
> > - SnapKV is a compression algorithm that uses voting and clustering mechanisms to determine important KV positions within a sequence. However, it overlooks the importance of different layers in the model. To address this limitation, we propose two potential integration strategies:
> > 1. **Sequential-First Integration**: First, apply SnapKV to identify and preserve valuable prompt tokens at the sequence level. Then, use SqueezeAttention to reallocate KV budgets across layers based on their importance.
> > 2. **Layer-First Integration**: Alternatively, SqueezeAttention can first reallocate KV budgets for each layer according to their importance. Subsequently, SnapKV can further compress the KV size within each layer based on a predefined proportion, ensuring a balanced optimization across both sequence and layer dimensions.
> > Furthermore, both algorithms focus on compressing KV size during the prefilling phase, which ensures that this combined approach is computationally efficient and feasible to implement in practice.
> > - PyramidKV is conceptually similar to SqueezeAttention, as both dynamically adjust KV cache sizes across layers. PyramidKV achieves this through pyramidal information funneling, optimizing KV allocation based on the assumed attention distribution across layers. However, integrating these two algorithms may prove challenging since both operate at the same level of optimization.
> > The key distinction lies in their design philosophy: PyramidKV is a standalone algorithm. While it demonstrates excellent performance in Llama and Mistral within its experiments, its generalizability to other models and datasets remains to be fully validated.
> > In contrast, SqueezeAttention is a combinatory framework designed to integrate with other sequence-wise KV compression methods. This design enhances its generalization capability by leveraging the strengths of diverse algorithms. For instance, if SqueezeAttention integrates effectively with SnapKV—which performs well on LWM-Text-Chat-1M—this not only validates SqueezeAttention’s adaptability but also highlights its potential utility in scenarios like LWM-Text-Chat-1M.
> > - KIVI focuses on quantization, reducing KV cache size by employing a 2 bit asymmetric quantization scheme for keys and values. By combining KIVI's quantization with SqueezeAttention's dynamic layer-level reallocation, we can achieve a two-pronged optimization:
> > Step 1: Use SqueezeAttention to allocate KV resources dynamically across layers based on their importance.
> > Step 2: Apply KIVI within each layer to further compress the allocated KV resources via quantization, ensuring maximum memory efficiency.
> > The benefits are obvious, the combination reduces overall memory footprint and computational overhead, especially in long-context tasks. However, it also face challenges, for example: KIVI introduces quantization-induced precision loss, SqueezeAttention must ensure that its reallocation does not amplify these effects.
> >
> > In SqueezeAttention, the integrated algorithms may be relatively simple; however, we believe we have successfully demonstrated the feasibility of integrating SqueezeAttention with other compression methods, yielding promising results. Extending this work to incorporate more complex algorithms would require further research and effort, which we consider a highly promising direction for future work.
> >
> > 3. In Table 3, it's a little bit unfair to compare the thoughput only with the full cache, since the KV cache evicted method is not the contribution of this work while the part of the thoughput improvements is achieved by the kv eviction, rather than the layer-wise strategy.
> >
> > **Answer:**
> >
> > | Mistral-7B | 1 | 32 | 64 | 128 | 224 |
> > | --- | --- | --- | --- | --- | --- |
> > | SqueezeAttention | 20.5 | 504.1 | 689.9 | 824.8 | 893.5 |
> > | Sliding Window | 20.6 | 404.5 | 512.2 | 587.8 | OOM |
> >
> > | LLama2-7B | 1 | 32 | 64 | 128 |
> > | --- | --- | --- | --- | --- |
> > | SqueezeAttention | 20.0 | 143.0 | 150.4 | 144.9 |
> > | StreamingLLM | 20.4 | 113.7 | 102.4 | OOM |
> >
> > We add two experiments to compare the throughputs of SqueezeAttention with Sliding Window and StreamingLLM under a set of batch sizes. Both experiments used an input length of 512 and an output length of 1024. We chose the compression hyper-parameters for each algorithm such that they could all achieve the best mode accuracy. The results show that our algorithm can obviously increase the throughput compared with those SOTA algorithms that only compress KV-cache from the sequence's dimension.
> >
> > Besides, due to space constraints in the paper, the above results are not included in the main text but are provided in the appendix

---

> > > ### Author Response · Authors · 2024-11-17
> > > **response to reviewer2BQ8 (part 3)**
> > >
> > > **Questions:**
> > >
> > > 1. Do G1,G2,G3 changes frequently across different samples? otherwise we can assign the layer-wise budget through a offline process.
> > >
> > > **Answer**:
> > >
> > > We conducted an extra experiment using two models and various datasets to determine whether the importance of different layers is an intrinsic property of the model.
> > >
> > > The table below displays the distribution of important layers for the Mistral model across three different datasets: Samsum (Few shot), TriviaQA (Single-document QA), and LCC (Code, Python/C#/Java).
> > >
> > > | Dataset | Samsum | TriviaQA | LCC |
> > > | --- | --- | --- | --- |
> > > | Important layers | 17 | 18 | 19 |
> > > | Unimportant layers | 15 | 14 | 13 |
> > >
> > > The next table shows the distribution of important layers for the LLama2-70B model across three different datasets: Xsum (Summarization), Samsum (Few shot), and LCC (Code, Python/C#/Java).
> > >
> > > | Dataset | Xsum | Samsum | LCC |
> > > | --- | --- | --- | --- |
> > > | Important layers | 17 | 21 | 18 |
> > > | Unimportant layers | 63 | 59 | 62 |
> > >
> > > From these tables, we can observe that there is a rough pattern regarding the layer’s group with task-specific fluctuations. We believe there exist some task-sensitive layers that may be classified into different groups with different tasks. Similarly, there are also some layers that are always important / unimportant. A detailed analysis of this phenomenon could be an interesting extension of this work. However, we would still recommend an adaptive way since it can precisely capture the importance of layers.
> > >
> > > The result above can also be found in the appendix of our paper.

---

> > > ### Comment · Reviewer_2BQ8 · 2024-11-25
> > > **Thanks for the responses**
> > >
> > > Thanks for your efforts in the rebuttal. The responses have addressed some of my concerns. However, I have a few follow-up questions:
> > >
> > > - Could you provide the detailed hyperparameters for each algorithm in the throughput comparison experiments, as well as information about the evaluated device (specifically, the GPU’s memory budget and other relevant specifications)?
> > >
> > > - Additionally, as discussed, PyramidKV is also a layer-wise KV cache compression strategy and should be included as a baseline method for comparison.
> > >
> > > - Lastly, I remain somewhat concerned about the KV cache eviction policy used. While H2O and StreamingLLM were considered, these approaches are slightly dated. Incorporating more recent methods, such as SnapKV or MInference, would provide a more comprehensive evaluation.

---

> > > > ### Author Response · Authors · 2024-12-02
> > > > **response to review 2BQ8**
> > > >
> > > > Q: Could you provide the detailed hyperparameters for each algorithm in the throughput comparison experiments, as well as information about the evaluated device (specifically, the GPU’s memory budget and other relevant specifications)?
> > > >
> > > > A:  Please refer to the table for the detailed settings of the throughput experiments. We’d like to highlight that to achieve the same level of accuracy, squeezeattention manages to compress the KV cache more aggressively because of the layer-wise budget adaptation, which leads to the higher throughput and larger batch size.
> > > >
> > > > | Algorithm | model | prompt len + output len | KV Budget  | p value | GPU specs | Max batch size | device_map |
> > > > | --- | --- | --- | --- | --- | --- | --- | --- |
> > > > | Squeezeattention | Mistral-7B | 512 + 1024 | 512*20%=102.4 | 0.3 |  8*A100GPUs 80GB HBM for each GPU NVLink enabled | 224 | auto |
> > > > | Sliding Window | Mistral-7B | 512 + 1024 | 512*40%=204.8 | none |  8*A100GPUs 80GB HBM for each GPU NVLink enabled | 128 | auto |
> > > > | Squeezeattention | LLama2-7B | 512 + 1024 | 512*30%=153.6 | 0.3 | 8*A100GPUs 80GB HBM for each GPU NVLink enabled | 128 | auto |
> > > > | StreamingLLM | LLama2-7B | 512 + 1024 | 512*60%=307.2 | none |  8*A100GPUs 80GB HBM for each GPU NVLink enabled | 64 | auto |
> > > >
> > > > Q: Additionally, as discussed, PyramidKV is also a layer-wise KV cache compression strategy and should be included as a baseline method for comparison.
> > > >
> > > >
> > > > Q: Lastly, I remain somewhat concerned about the KV cache eviction policy used. While H2O and StreamingLLM were considered, these approaches are slightly dated. Incorporating more recent methods, such as SnapKV or MInference, would provide a more comprehensive evaluation.
> > > >
> > > > Thanks for your suggestion! We have conducted more experiments to compare with PyramidKV and SnapKV. Let’s first have a quite recap of these algorithms.
> > > >
> > > > - SnapKV is conceptually similar with H2O, which selects the “important” tokens out of the sequence and evicts the KV cache of the rest part. Note that both SnapKV and H2O assume each layer has the same KV cache budget.
> > > > - Based on SnapKV, PyramidKV adjusts the cache budget for each layer by an arithmetic sequence.
> > > >
> > > > Since SqueezeAttention is designed for compatibility, we can easily integrate SnapKV. We follow the experiment settings of SnapKV and PyramidKV: The model is Mistral-7B-Instruct-v0.2. We set the hyperparameter of SqueezeAttention (p) to 0.7 and standardized the prompt size across all three methods to 2048. Due to time constraints, we only tested a subset of datasets from LongBench (one dataset per Task Type). We used the results reported in the SnapKV and PyramidKV papers.
> > > >
> > > > |  | hotpotqa | gov_report | triviaqa | lcc |
> > > > | --- | --- | --- | --- | --- |
> > > > | Ours + SnapKV | **42.42** | 29.0 | 86.33 | 53.52 |
> > > > | Ours + H2O | 38.06 | **30.88** | **87.72** | 53.41 |
> > > > | PyramidKV | 42.26 | 26.60 | 86.25 | 53.12 |
> > > > | SnapKV(same KV budget for each layer) | 41.71 | 28.81 | 86.27 | **55.93** |
> > > >
> > > > The results show that in most cases, SequenceAttention could outperform SnapKV and PyramidKV, thanks to our great compatibility. We believe this experiment also reveals that currently there is no “**one-for-all”** KV cache compression strategy that always work best. Different models and tasks react quite differently to those approximation methods. Therefore, the openness and compatibility of our algorithm make it applicable to a broader range of tasks.
> > > >
> > > > Besides, there is another strength we have over PyramidKV. Defined by an arithmetic sequence, PyramidKV automatically assumes that the deeper layers should cache less KV embeddings, which, although holds for some tasks of llama and mistral, is not always true given our observation. For example, the last layer of Falcon-7B has a sudden reversal in embedding cosine similarity, indicating the great importance of the last layers. [1] also observed that “…for the initial and final layers, they have more attention heads assigned to the full KV cache, indicating attention heads in these layers are likely to attend to all tokens...” in Llama 165B on GSM8k dataset. Whereas, our method is able to detect the importance of each layer adaptively on-the-fly given the model and task.
> > > >
> > > > [1] Ge, Suyu, et al. "Model tells you what to discard: Adaptive kv cache compression for llms."  ICLR 2024.

---

> > > > ### Author Response · Authors · 2024-12-04
> > > > **Thanks for the suggestions**
> > > >
> > > > Dear Reviewer 2BQ8,
> > > >
> > > > Thanks again for your suggestions regarding the comparisons with more recent related works. We have provided additional information for your consideration:
> > > > 1. Detailed hyperparameters regarding the additional throughput experiments.
> > > > 2. Integrate SnapKV into SequeezeAttention and evaluate it on the LongBench dataset.
> > > > 3. Comparison with PyramidKV on the LongBench dataset.
> > > > 4. Analysis of the experiment results.
> > > >
> > > > Please let us know if there are any further questions or suggestions. Thanks.

---

> ### Author Response · Authors · 2024-11-23
> **response to reviewer2BQ8**
>
> Once again, thank you for reviewing the paper. We think we have solved your problem as much as possible in rebuttal, if you have any further questions, please do not hesitate to contact us. If possible, we would like to thank you for reconsidering to improve your score.

---

### Official Review · Reviewer_ARNg · 2024-11-10

**Soundness:** 3
**Presentation:** 3
**Contribution:** 3
**Rating:** 6
**Confidence:** 4

**Summary:**

This paper proposes SqueezeAttention, a KV-Cache management algorithm that can be combined with KV-Cache eviction policies to further reduce memory footprint and improve throughput. SqueezeAttention allocates size budgets for the KV-Cache of different layers by utilizing statistics on the importance of the attention layers. Specifically, SqueezeAttention first computes the cosine similarity between the activations before and after each attention layer.  Based on this similarity, the layers are then categorized into two groups and their KV budgets adjusted accordingly. SqueezeAttention achieves around 30% to 70% memory reductions and up to 2.2 × of throughput improvements in a wide range of LLMs and benchmarks.

**Strengths:**

1. The method can augment other KV-Cache eviction policies, which will benefit the research community.
2. The algorithm is clearly presented and the method's effectiveness has strong experiment evidence.

**Weaknesses:**

1. There's little analysis of the reason for performance improvement as shown in Figure 3. Some hypothesis or statistics analyses could give readers a deeper understanding of the algorithm.
2. The memory usage of Figure 4 is not clearly explained. What tensors are counted in the PyTorch Profiler? Besides, why does LLama2-70B consume a similar amount of memory to Mistral-7B?

**Questions:**

1. What inference framework is used for the memory and throughput experiments? Is SqueezeAttention compatible with current inference memory optimization like vllm[1]?

[1] Kwon, W., Li, Z., Zhuang, S., Sheng, Y., Zheng, L., Yu, C.H., Gonzalez, J., Zhang, H. and Stoica, I., 2023, October. Efficient memory management for large language model serving with pagedattention. In Proceedings of the 29th Symposium on Operating Systems Principles (pp. 611-626).

---

> ### Author Response · Authors · 2024-11-16
> **Response to reviewer ARNg**
>
> Dear reviewer, thank you very much for your comments and professional advice. Based on your suggestions, we clarify some ambiguous parts of the paper and would like to provide the details as follows:
>
> **Weaknesses**:
>
> 1. There's little analysis of the reason for performance improvement as shown in Figure 3. Some hypothesis or statistics analyses could give readers a deeper understanding of the algorithm.
>
> **Answer:**
>
> Thanks for your advice and we’d like to describe Fig. 3 with more details. Basically we can interpret the Fig. 3 in three steps: 1) The Full Cache line represent the ideal model performance since it simply caches all tokens’ KV embeddings for all layers (default self-attention algorithm). 2) Then we apply three representative sequence-wise KV sparsification algorithms to evict tokens with an identical strategy and budget for all layers, we can see as the total KV budget goes down, the model performance drops accordingly. Note that for each model and task, we choose the best sequence-wise algorithm to represent the best baseline. 3) Finally, we use SqueezeAttention to adjust the cache budget for each layer based on the best baseline in step 2). As we can see, for a given KV budget, SqueezeAttention almost always achieve better performance than the best baseline. In other words, to reach a given performance, SqueezeAttention always requires less KV budget than best baseline. The reason of performance improvement is that SqueezeAttention optimizes the distribution of KV cache budgets over layers by prioritizing the important layers, instead of allocating same budget to all layers like the baseline algorithms do.
>
> 2. The memory usage of Figure 4 is not clearly explained. What tensors are counted in the PyTorch Profiler? Besides, why does LLama2-70B consume a similar amount of memory to Mistral-7B?
>
> **Answer:**
>
> In our memory and time efficiency experiments, we used `with profiler.record_function("model_inference"):` to capture memory and time consumption during the inference process. The profiling results that it is the KV cache that dominates the memory cost of inference, which aligns with our assumption.
>
> The tensors recorded primarily include the KV-cache embeddings and activation tensors generated in the forward phase, but **exclude** model parameters. LLama2-70B has more layers (80) than Mistral-7B (32), which leads to more KV embeddings and activations. However, Mistral-7B has longer context length (32k) than LLama2-70B (4k), which leads to more tokens cached. Therefore, they turn out to consume similar memory overall.
>
> **Questions:**
>
> 1. What inference framework is used for the memory and throughput experiments? Is SqueezeAttention compatible with current inference memory optimization like vllm[1]?
>
> **Answer:**
>
> For the memory and throughput experiments, we used the **Hugging Face Transformers** framework with **Flash Attention** enabled. In theory, SqueezeAttention, as an inference algorithm, is compatible with most inference frameworks, including vLLM. However, optimizations implemented by frameworks like vLLM and DeepSpeed-Fastgen operate at the kernel level, which limits their flexibility. These frameworks are designed to optimize computation and I/O operations in a highly specialized way to maximize inference speed.
>
> These optimization frameworks, while effective, make it challenging to integrate new algorithms, as it requires significant modifications. For instance, according to Mistral’s blog, integrating the sliding window algorithm into vLLM has required assistance from both the vLLM and FlashAttention teams. We are currently working on integrating SqueezeAttention with both DeepSpeed and vLLM, and we aim to have this integration ready by the camera-ready version of the paper.

---

> > ### Comment · Reviewer_ARNg · 2024-11-16
> >
> > Thanks for your response. However, the memory measurement still seems incorrect to me. Let's take LLama2-70B as an example. It has 80 layers. The hidden size is 8192. As suggested by your response, the sequence length is 4096 (4K). Let's assume fp16. The total size of the full KV-Cache is 80 * 8192 * 4096 * 2B * 2 = 10GB. However, it's only 5.73GB in Figure 4 (a). Could you further clarify the measurement?

---

> > > ### Author Response · Authors · 2024-11-17
> > >
> > > Thank you for pointing out the discrepancy in the memory measurements. Your analysis is correct, and the difference lies primarily in the dataset configuration used in our experiments. Allow me to clarify our experimental settings further.
> > >
> > > In our experiments, we used datasets with varying average sequence lengths, as detailed in **Table 1** of our paper. For the LLaMA2-70B model, the memory experiments were conducted on the **XSUM dataset**, which has an **average sequence length of 2000 tokens**. When recalculating the KV cache size based on this average length, the theoretical memory usage aligns closely with the observed results. Specifically:
> > >
> > > Memory usage=80×8192×2000×2B×2≈5GB
> > >
> > > This result is consistent with the reported memory usage in Figure 4(a). The small differences arise due to:
> > >
> > > 1. **Variations in input lengths:** While the average length is 2000, specific samples may have longer input lengths, especially after tokenization.
> > > 2. **Unaccounted memory components:** The theoretical calculation does not include additional memory usage, such as activation memory and other runtime overheads, which contribute to the slight deviation.
> > >
> > > We intentionally chose to use the same datasets for the memory experiments as those in the accuracy experiments because our goal was to measure the memory savings achieved by **SqueezeAttention** under the same accuracy conditions. By maintaining consistency in datasets, we can better evaluate how much memory our method can save without compromising accuracy. This approach ensures the practical relevance of our results and highlights the efficiency of SqueezeAttention in reducing memory usage.
> > >
> > > Additionally, as shown in **Table 2**, our memory experiments align with the findings from the accuracy experiments. For example, LLaMA2-70B with SqueezeAttention achieves comparable accuracy to the full KV cache while utilizing only **30% of the total cache** compared to **40% without SqueezeAttention**. This result is consistent with **Table 4**, demonstrating the efficiency of our method in both memory savings and accuracy retention.
> > >
> > > For reference, similar theoretical calculations for other models also match closely with experimental results:
> > >
> > > - **Mistral:** 32×4096×6258×2B×2=3.28GB
> > > - **GPT-NeoX:** 44×6144×2000×2B×2=2.1GB
> > >
> > > We hope this explanation resolves the concerns regarding memory measurement and provides clarity on the experimental setup and results. Thank you for your valuable feedback.

---

> ### Author Response · Authors · 2024-11-23
> **Response to reviewer ARNg**
>
> Dear reviewer:
>
> Thank you again for your valuable feedback. This is just a kind reminder that we have  addressed the concerns you raised and further clarified the experimental setup in our response.
>
> We would greatly appreciate it if you could reconsider the assessment. Please don't hesitate to reach out if you have any further questions.
>
> Thank you for your time and support!

---

> > ### Comment · Reviewer_ARNg · 2024-12-01
> >
> > Thank the authors for the response. My concerns have been resolved and I will keep the score.

---

### Meta-Review · Area_Chair_Gg1h · 2024-12-20

**Metareview:**

The paper presents a practical approach of layer-wise dynamic allocation for KV compression,  based on their importance determined by the cosine similarity of embeddings before and after self-attention layers. The method is compatible with other sequence-based compression algorithms, augmenting their performance by optimizing layer-level cache budgets. In their experiment, the method achieves 30%-70% memory reduction and up to 2.2× throughput improvement across diverse models (i.e., Llama2, Mistral, Falcon, OPT, GPT-Neox, etc) combined with 3 representative sequence-wise compression algorithms (i.e. H2O, Sliding window and Streaming LLM).

Reviewers generally praised the paper for its comprehensive evaluation and practical benefits. There is initially a missing comparison with layer-adaptive compression methods like PyramidKV which weakens the experimental design, but the authors conducted experiments during the rebuttal period that provided a comparison. Another shortcoming comes from fixed three-group clustering and fixed design of compression ratio which in principle may limit its adaptability for broader applications. However, the concern is alleviated by the extensive experiments that demonstrated the strong performance of their method.

Overall, the paper represents a solid study of a layer-adaptive KV compression technique, which could be of great practical interest. The paper is relatively weak in terms of novelty as layer-adaptive strategies have been explored before in PyramidKV, and has a relatively limited scope since it does not have much broader implications beyond KV compression. Therefore I recommend an acceptance for a poster presentation.

**Additional Comments On Reviewer Discussion:**

Concern: Lack of comparisons with PyramidKV, SnapKV, and other recent methods.

Response: Authors added preliminary experiments comparing SqueezeAttention with PyramidKV and SnapKV, showing competitive performance.

Concern: Fixed three-group clustering limits adaptability.

Response: Authors argued this approach is sufficient for most observed patterns and reduces complexity.

Concern: Overhead analysis was insufficient.

Response: Additional experiments showed minimal computational cost compared to baseline methods.

Concern: Does not extend to dynamic KV-cache adjustments during decoding.

Response: Authors clarified that real-time adjustment would incur significant computational costs, making it impractical.

---

### Decision · Program_Chairs · 2025-01-22

Accept (Poster)